# School-Based Exercise and Life Style Motivation Intervention (SEAL.MI) on Adolescent’s Cardiovascular Risk Factors and Academic Performance: Catch Them Young

**DOI:** 10.3390/healthcare9111549

**Published:** 2021-11-13

**Authors:** Premalatha Paulsamy, Kalaiselvi Periannan, Vigneshwaran Easwaran, Noohu Abdulla Khan, Vani Manoharan, Krishnaraju Venkatesan, Absar Ahmed Qureshi, Kousalya Prabahar, Geetha Kandasamy, Rajalakshimi Vasudevan, Kumarappan Chidambaram, Ester Mary Pappiya, Kumar Venkatesan, Pranave Sethuraj

**Affiliations:** 1College of Nursing, Mahalah Branch for Girls King Khalid University, Khamis Mushayt 61421, Saudi Arabia; pponnuthai@kku.edu.sa; 2Oxford School of Nursing & Midwifery, Faculty of Health and Life Sciences, Oxford Brookes University, Oxford OX3 0FL, UK; kperiannan@brookes.ac.uk; 3Department of Clinical Pharmacy, College of Pharmacy, King Khalid University, Abha 62529, Saudi Arabia; vickku.e@gmail.com (V.E.); nakhan1786@gmail.com (N.A.K.); 4Georgia CTSA, Emory University Hospital, Atlanta, GA 30078, USA; vani.manoharan@emoryheslthcare.org; 5Department of Pharmacology, College of Pharmacy, King Khalid University, Abha 62529, Saudi Arabia; aqureshi@kku.edu.sa (A.A.Q.); glakshmi@kku.edu.sa (G.K.); raja@kku.edu.sa (R.V.); kumarappan@kku.edu.sa (K.C.); 6Department of Pharmacy Practice, Faculty of Pharmacy, University of Tabuk, Tabuk 71491, Saudi Arabia; kgopal@ut.edu.sa; 7Regional Nursing Administration, Directorate of General Health Affair, Ministry of Health, Najran 21431, Saudi Arabia; easterbala@gmail.com; 8Department of Pharmaceutical Chemistry, College of Pharmacy, King Khalid University, Abha 62529, Saudi Arabia; kumarve@kku.edu.sa; 9Vee Care College of Nursing, The TN MGR Medical University, Chennai 600095, India; spranave98@gmail.com

**Keywords:** blood pressure, rope exercise, physical activity, lifestyle modification, adolescents, BMI, hypertension, cardiovascular risk factors, academic performance

## Abstract

There are shreds of evidence of shared biological mechanisms between obesity and hypertension during childhood intoadulthood, and loads of research literature has proven that it will profoundly cost nations’ economies and health if neglected. The prevention and early diagnosis of cardiovascular risk factors such as overweight and hypertension is an essential strategy for control, effective treatment and prevention of its’ complications. The study aims to assess the effect of school-based Exercise and Lifestyle Motivation Intervention (SEAL-MI) on adolescents’ cardiovascular risk factors and academic performance. An experimental study was conducted among 1005 adolescents—520 and 485 were randomly selected for the control and study groups, respectively.A structured interview questionnaire was used to collect demographic details and data related to dietary habits, physical activity, sleep qualityand academic performance. The study group adolescents were given the SEAL-MI for six months, including a school-based rope exercise for 45 min per day for 5 days a week and a motivation intervention related to dietary habits, physical activity, and sleep. Post tests-1 and 2 were done after 3 and 6 months of intervention.The prevalence of overweight among adolescents was 28.73%, and prehypertension was 9.26%. Among overweight adolescents, the prevalence of prehypertension was found to be very high (32.25%). There was a significant weight reduction in post-intervention B.P. (*p* = 0.000) and improvement in dietary habits, physical activity, sleep (*p* = 0.000), and academic performance. A significant positive correlation was found between BMI and SBP (*p* = 0.000) and BMI and academic performance (*p* = 0.003). The linear regression analyses revealed that the gender (ß: 0.47, 95% CI: 0.39, 0.81), age (ß: 0.39, 95% CI: 0.17, 0.46), family income (ß: 0.2, 95% CI: 0.41, 0.5), residence (ß: 0.19, 95% CI: 0.01, 0.27), and type of family (ß: 0.25, 95% CI: 0.39, 0.02) had the strongest correlation with the BMI of the adolescents. Additionally, Mother’s education (ß: 0.35, 95% CI: 0.18, 0.59) had the strongest correlation with the SBP of the adolescents. In contrast, the DBP was negatively persuaded by age (ß: −0.36, 95% CI: 1.54, 0.29) and gender (ß: −0.26, 95% CI: 1.34, 0.12) of the adolescents. Regular practice of rope exercise and lifestyle modification such as diet, physical activity, and quality sleep among adolescents prevent and control childhood CVD risk factors such asoverweight and hypertension. The SEAL-MI may lead to age-appropriate development of adolescents as well as improve their academic performance and quality of life. Giving importance to adolescents from urban habitats, affluent, nuclear families, and catching them young will change the disease burden significantly.

## 1. Introduction

The world of tomorrow will inherit the adolescents of today. Developing the full potential of adolescents and helping them to survive with good health is essential. Nevertheless, due to the constant development of economic level and corresponding lifestyle changes, we are facing a gradual increase in weight with the younger-age trend. In the United States, according to the Centre for Disease Control and Prevention (CDC), approximately 17% (i.e., 12.5 million) of children and adolescents aged 2–19 years are obese, with the obesity prevalence in these groups having almost tripled since 1980 [1]. There has also been an increase in the global age-standardized mean body mass index (BMI) of children and adolescents aged 5–19 over the last four decades, from 17.2 to 18.6 kg/m^2^ in girls (an increase of 0.32 kg/m^2^ in one decade) and from 16.8 to 18.5 kg/m^2^ in boys (an increase of 0.4 kg/m^2^ in one decade) [1].

Obesity causes various disorders, and it is of great public health concern across the world. According to the World Health Organization (WHO), childhood obesity has been strongly associated with a higher chance of premature death and disability in adulthood [2,3]. Central adiposity, defined as fat mass accumulation around the abdomen, is highly correlated with cardiovascular risks such as hypertension, elevated plasma lipid concentrations, lipoprotein concentrations, C-reactive protein(CRP)levels, insulin resistance (I.R.) [3], and changes within the vasculature such as increased arterial stiffness [4] and blood pressure (B.P.) [5] in children and adolescents [6].

The behavioral risk factors such as an unhealthy diet are estimated to be related to about half of hypertension (about 30% related to increased salt consumption, and about 20% related to low dietary potassium, low fruit and vegetables). Insufficient physical activity is the fourth leading risk factor for mortality. Physical inactivity is related to about 20% of hypertension, and obesity is also related to about 30%. More than two-thirds of the adolescents are physically inactive in India, as per WHO standards. Any preventive and promotive health interventions targeted in this age will be relatively cost-effective yet yield a lifetime gain, not only for individuals, also for societies and economies. Additionally, school-based childhood interventions are among the most cost-effective in saving the Disability Adjusted Life Years (DALYs) in their later life.

Captivatingly, adolescents who have a low level of physical activity are more prone to be obese, depressed and more likely to have a lower grade point average (GPA) [7]. Academic performance, in general, is considered to be associated with cognitive and memory functions [8]. Known the negative relationship of obesity with cognitive and memory functions, overweight or obese might negatively impact adolescents’academic achievement [9]. Indeed, a high body mass index (BMI) has been linked with negative alterations in brain structure and an increased risk of Alzheimer’s disease [10].

Further, depressed adolescents are more likely to develop and sustain obesity during their adolescence. Understanding the shared biological and social variables that link depressed mood and obesity could help with mutually prevention and therapy of both disorders. Depression in adolescents has also been linked to a higher body mass index in adulthood, though it is uncertain whether depression promotes obesity or obesity causes depression [11]. Hence, there must be an intervention that involves adolescents participating in a daily schedule of school-basedexercise activities and healthy eating in order to maintain the weight loss and B.P. reduction, as there will be many distractions like food advertising and marketingand community/family environment for the adolescents to overlook the schedule or lose follow up. Participants in this type of intervention may thus benefit from continued support after returning home involving family as well, to minimize attrition.

For these reasons, school-based healthy lifestyle curriculum development is the need of the hour for adolescents as there is relative evidence that physical activity elevates adolescents’mood, improves memory, cognitive function, and prevents CVD risk factors. Additionally, school is an easily accessible and affordable ambience to adolescents, and teachers play a crucial role in children’s physical, psychological, social, and emotional development. Therefore, the main objectives of this study were to estimate the prevalence of overweight and hypertension among adolescents and to assess the effect of school-based Exercise and Lifestyle Motivation Intervention (SEAL-MI) on adolescents’ cardiovascular risk factors and Academic performance.

## 2. Methods

### 2.1. Design

This experimental study with before and after control group designwas conducted in the randomly selected high schools of Tiruvallur district, Tamilnadu, India.

### 2.2. Population and Setting

Adolescents with overweight and increased blood pressure were the target population of the study. The Tiruvallur educational District, Tamilnadu, India, had 2263 functioning Schools in both rural and urban zone, and data collection took place in the chosen high schools.

### 2.3. Sample Size and Sampling Process

The sample size was calculated based on the power analysis taking 95% confidence level and 80% power of the study with α = 5%, β = 20%. The sample size was estimated with the improvement of 20% gain from baseline 50% with 80% power of the study, 5% allowable error, and 10% dropout rate. The estimated sample size was rounded as 500 in each control and intervention group. An updated list of all public schools was used as the sampling frame, and stratified sampling method was used. The schools were stratified proportionately according to urban/rural distribution; after ensuring matching, eight rural zones and eight urban wards were selected, and they were randomly allotted into intervention and control groups using the lottery method. In the same way, two schools (for intervention and control group) from rural and urban wards were selected in the second stage. From the 16 selected schools from each urban and rural area, all the adolescents aged 13–14 years, who were willing to participate in the study, were screened for hypertension and overweight as a first phase (Figure 1).

After obtaining approval from the Institutional Ethical Committee, the baseline information was assessed by the modified “WHO STEPS” instrument to monitor chronic disease risk factors [12]. The adolescents who fell in ≥85th percentile according to the WHO chart on BMI for age (Boys and Girls) and have ≥90th Systolic/Diastolic Percentile according to National High Blood Pressure Education Program (NHBPEP) Working Group on Adolescents and National High Blood Pressure Education Program (NHLBI), were included for the study.

The adolescents diagnosed with pathological causes of hypertension such as renal diseases, cardiac conditions with any other co-morbidities and who were already on medication for overweight and hypertension were excluded. The adolescents who had significant complications associated with medical conditions, learning disabilities, breathing, hormonal and musculoskeletal limitations, on an exercise program, and unable to perform moderate to vigorous-intensity exercises were excluded.

Initially, 12,895 adolescents from 16 rural and 16 urban schools were screened for obesity and hypertension in the first phase. Among them, 3705 were overweight, and 1195 were identified with both overweight and prehypertensive. Based on the inclusive and exclusive criteria, 1005 adolescents with overweight and prehypertensive were randomized and allotted by lottery method in the study (500) and control (485) group.

### 2.4. Tools/Instruments

Section A: Screening Tool, which was used to screen overweight adolescents with increased blood pressure, measuring height, weight, calculated body mass index, and blood pressure.

Section B: Assessment tool was used to collect the demographic, lifestyle, and bio-physiological variables. It consists of three parts. Part A: Demographic variables of the adolescents consists of age, gender, religion, level of education, birth order in the family, number of siblings, number of persons in the family, monthly income of the family, mother and father’s education and working status, type of family, and nature of residence as well as family history of overweight and any other chronic illnesses. Part B: Lifestyle Variables, which was assessed by the modified tool, “WHO STEPS” instrument, used for the surveillance of chronic disease risk factors. It included dietary habits and physical activity. Part C: BEARS sleep screening test used to assess the adequacy of sleep among adolescents. Part D: Bio- Physiological variables include height, weight, Body Mass Index (BMI), and blood pressure.

Using the BMI Percentile Chart for boys and girls by WHO (2007), if a child had a percentile ranking of 85th or greater was considered overweight. Interpretation of BP was done by following the 2004 guidelines produced by the National High Blood Pressure Education Working Group on High Blood Pressure in Children and Adolescents of the National Heart, Lung and Blood Institute blood pressure cut-points classified based on age, gender, and height percentile-specific systolic blood pressure (SBP) and diastolic blood pressure (DBP) levels. Age and height adjusted SBP and DBP was estimated for each participant according to the 4th report on Diagnosis, Evaluation and treatment of High B.P. in children and adolescents (2004) guidelines, and blood pressure percentiles were calculated for each child based on the expected blood pressure. In adolescents, the normal range of B.P. is determined by body size and age. This approach avoids misclassifying adolescents who are very tall or very short.

The height percentile was determined using the newly revised WHO Growth Charts to use the tables in a clinical setting. The adolescents’ measured SBP and DBP are compared with the numbers provided in the table (boys or girls) according to the adolescent’s age and height percentile. The adolescent is normotensive if the B.P. is below the 90th percentile. If the B.P. is equal to or above the 90th percentile, the B.P. measurement was repeated at that visit to verify elevated B.P. The B.P. measurements between the 90th and 95th percentiles indicate prehypertension and warrant reassessment and consideration of other risk factors. In addition, if an adolescent’s B.P. is greater than 120/80 mmHg, the adolescent was considered prehypertensive even if this value is less than the 90th percentile. This B.P. level typically occurs for SBP at age 12 years and DBP at age 16 years. If the adolescent’s B.P. (systolic or diastolic) is at or above the 95th percentile, the adolescent may be hypertensive, and the measurement was repeated on at least two different occasions to confirm the diagnosis.

Using the blood pressure tables: 1. Use the standard height charts to determine the height percentile, 2. Measure and record the child’s SBP and DBP, 3. Use the correct gender table for SBP and DBP, 4. Find the adolescent’s age on the left side of the table. Follow the age row horizontally across the table to the intersection of the line for the height percentile (vertical column), and 5. Find the 50th, 90th, 95th, and 99th percentiles for SBP in the left columns and DBP in the right columns.

B.P. less than the 90th percentile is normal.B.P. between the 90th and 95th percentile is prehypertension. In adolescents, B.P. equal to or exceeding 120/80 mmHg is prehypertension, even if this figure is less than the 90th percentile.B.P. greater than the 95th percentile may be hypertension.

If the B.P. is greater than the 90th percentile, the B.P. was repeated twice at the same office visit, and an average SBP and DBP was used. If the B.P. is greater than the 95th percentile, B.P. was staged. If Stage 1 (95th percentile to the 99th percentile plus 5 mmHg), B.P. measurements were repeated on two more occasions. If B.P. was Stage 2 (>99th percentile plus 5 mmHg), a prompt referral was made for evaluation and treatment. If the adolescent was symptomatic, an immediate referral was made, and treatment was advised.

### 2.5. Intervention

The school-based Exercise and Life Style Motivation Intervention (SEAL-MI) incorporated planned teaching on knowledge regarding prevention and control of blood pressure, overweight, dietary modifications, rope exercise program, and activities to enhance sleep quality. The investigators administered the intervention, designed to change adolescents’ BMI and BP for 6 months. The teaching was given at the beginning of the intervention, and the rope exercise program was carried out for 6 months with a frequency of five times a week in the school after the school hours under the supervision of physical education teachers. The physical education teachers (PET) were given intensive training for 3 days on the rope exercise program, physical activity monitoring tools, exercise intensity, and calculations of maximum heart rate (MHR). A WhatsApp group was maintained among PETs and the researchers for instant problem resolution if any. Before implementing the rope exercise program, a no harm certificate was received from a Paediatrician, a Physical therapist and a Clinical Nutritionist for ethical reasons.

During the rope exercise program, participants warmed up for 5 min and completed eight cycles of rope jumping exercise. Each cycle consisted of rope jumping for 2 min followed by 2 min rest which lasted for around 30 min, followed by relaxation exercise for 5 min. To maintain the exercise intensity, changes in heart rate was observed. In the current study, the participant’s mean age was 13 years 2 months, and the estimated MHR employed was 210 bpm. Therefore, the MHR giving rise to a change in heart rate (as defined by aerobic work) is 210 × 60% MHR = 126 bpm. The heart rate of every child was recorded daily at the end of each 2-minute cycle of rope jumping, before and after warm-up to keep up the required advantage of the exercise and a diary was maintained for each child by the physical education teachers who have been trained for the same. The compliance was ensured through phone calls, WhatsApp and weekly direct visits to the intervention group adolescents by one investigator on rotation. The investigators maintained a 6-month physical activity, diet and sleep record with the help of physical education teachers for rope exercise and parents for diet and sleep patterns to monitor the progress and compliance. After 3 and 6 months, post-interventional data were collected.

### 2.6. Data Collection Tools/Instruments

The primary dependent variables were the students’ bio-physiological variables, including height, weight, Body Mass Index (BMI) and blood pressure. Adopting the BMI Percentile Chart for boys and girls [12,13], if a child has a percentile ranking of 85th or greater was considered overweight. Age and height adjusted systolic blood pressure (SBP) and diastolic blood pressure (DBP) was estimated for each participant [14,15]. The blood pressure (B.P.), which was taken during the leisure time of the children such as lunch break, in a comfortable place assigned for that by using a mercury sphygmomanometer as per the recommendations of American Heart Association. Weekly calibration of the instruments was done. The child was normotensive if the B.P. was below the 90th percentile. If the B.P. was equal to or above the 90th percentile, the B.P. measurement was repeated three times with adequate rest to verify and confirm an elevated B.P. The tool’s reliability was assessed using Cronbach Alpha method and Test-retest methods (*r* = 0.92).

To assess the academic performance of adolescents, their self-reported academic performance was calculated by a question: “In the past 12 months, how has your average academic performance been?”. The answers were verified with their previous year school academic records, and the available responses were scored with a 5-point rating scale including 5 score for very good (average score above 80%) and 1 score for very poor (average score below 35%), respectively.

### 2.7. Ethical Consideration

Written consent was obtained from the Director, Directorate of Education, Tiruvallur District and Head of the Institution of all the schools selected for the study. Additionally, written consent from the parents and assent from the adolescents was obtained. The Institutional Ethical Committee approval was obtained from Meenakshi Academy of Higher Education and Research, Chennai.

### 2.8. Statistical Analysis

Bivariate Analysis, Correlation Coefficient, Chi-square test, paired, unpaired to test, repeated-measures ANOVA, multiple regression analysis was performed to compare scores over a period of time, within and between groups to determine the effect, correlate and associate the variables. To compare groups, the ‘*t*’ test for continuous variables and chi-square test for categorical variables was used. Repeated measures ANOVA were performed to compare scores over a period of time. Multiple regression analysis was completedto assess the strength of the relationship between outcome variables (BMI and BP) and the predictor variables (demographic characteristics). All results from this study were presented as mean ± standard deviation. Statistical significance was set at *p* < 0.05 and the data were entered into an Excel sheet and analyzed through Statistical Package for Social Science/PC^+^Ver.17.

## 3. Results

In the present study, a total of 12,895 adolescents aged between 13 and 14 years were screened by our team of investigators for overweight and hypertension. Among these adolescents, 3705 adolescents (28.73%) were found to be overweight, and 1195 adolescents (32.25%) were found to be overweight and prehypertensive.

Among the 1195 overweight and prehypertension, 1005 adolescents and their parents who were willing to participate in the study were selected randomly for the intervention (520) and control (485) group by lottery method. Considering the gender, approximately equal number of the sample, 231 (44.4%) boys and 285 (55.6%) girls in the study, as well as 281 (57.9%) girls and 204 (42.06%) boys in the control group, participated in the project. At the end of the Post test—1 & 2, only 500 and 485 adolescents were in the intervention and reference group, respectively, and the attrition rate was 3.85%. It was due to various reasons such as changing school or residence, dropping out from school, etc. There was homogeneity among selected samples in both the groups tested through “Test of Goodness of Fit”.

In the post test-1, 145 (29%) adolescents from the study group became normal weight which was statistically significant (*p* = 0.000). After six months of intervention, 405 (81%) study group adolescents became normal weight, none of them was obese, and only 95 (19%) adolescents were overweight, which was statistically significant (*p* = 0.000). The comparison of blood pressure status for study and control group adolescents showed that 105 (20.2%) study group adolescents become normotensive in the post test-1. In the post test-2, 347 (69.4%) study group adolescents became normal, which were statistically significant (*p* = 0.000), whereas only 39 (8.3%) of the control group became normotensive after 6 months. 

The comparison of BMI status among the study and control groups showed a significant difference in the study group between different study periods at *p* = 0.036 level as per repeated ANOVA. In contrast, the control group did not show any statistical difference in BMI between the study periods (Table 1). The effective score for BMI between study and control group compared by independent “*t*” test showed that though it was not significant, the difference found during both study periods confirms that the SEAL-MI was highly effective in reducing the BMI among study group as well as it demands the long-term intervention with appropriate reinforcement. Additionally, the significance level at 3 and 6 months indicates that it was easy to reduce BMI initially and to sustain the reduction was the more significant challenge.

Table 2 depicted that in the intervention group, the mean differed SBP is 0.27 with SD of 0.96 in pretest to post-test-1 and 1.57 with SD of 1.53 in pretest to post test-2, i.e., after 6months of intervention which were significant (*p* = 0.000). In contrast, the control group did not show any significant difference after 3 months but showed a significant difference after 6months (*p* = 0.03) due to the time effect. Similarly, the mean differed DBP was 0.09 with SD of 0.34 in pretest to post test-1 and 0.69 with SD of 1.26 in pretest to post test-2, which confirms that the intervention should be long term to bring desired change in DBP. Additionally, this confirms that the SEAL-MI was highly effective in reducing the B.P. among study group adolescents (Figure 2 and Figure 3).

Regarding the dietary habits, physical activity, and sleep quality based on BEARS Sleep Scale (B = bedtime problems, E = excessive daytime sleepiness, A = awakenings during the night, R = regularity and duration of sleep, and S = snoring), there was a significant difference within the study group at *p* = 0.000 level after 6months of intervention (Table 3).

Regarding association between BMI and systolic BP and demographic characteristics of study group, the ANOVA test findings in Table 4 revealed that there was statistically significant association found between the demographic variables such as gender (*p* = 0.004) (*p* = 0.000), education (*p* = 0.000) (*P* = 0.000), number of family members (*p* = 0.006) (*p* = 0.003), type of family (*p* = 0.035) (*p* = 0.002), family history of obesity and chronic illness (*p* = 0.036) (*p* = 0.031), respectively, with BMI and systolic BP of the adolescents in study group (Table 4).

The linear regression analyses (Table 5) revealed that the gender (ß: 0.47, 95% CI: 0.39, 0.81), age (ß: 0.39, 95% CI: 0.17, 0.46), family income (ß: 0.2, 95% CI: 0.41, 0.5), residence (ß: 0.19, 95% CI: 0.01, 0.27) and type of family(ß: 0.25, 95% CI: 0.39, 0.02) had the strongest correlate with the BMI of the adolescents. This suggests that gender, age, family income, residence, and family type strongly influence adolescents’ BMI status in the descending order.

The linear regression analyses (Table 6) revealed that the age (ß: 0.59, 95% CI: 0.7, 1.42), Mother’s education (ß: 0.35, 95% CI: 0.18, 0.59) and residence (ß: 0.26, 95% CI: 0.11, 0.73) had the strongest correlate with the SBP of the adolescents. This suggests that age, Mother’s education status, and place of residence strongly influence the SBP status of adolescents in descending order. In contrast, the DBP was negatively persuaded by age (ß: −0.36, 95% CI: 1.54, 0.29) and gender (ß: −0.26, 95% CI: 1.34, 0.12) of the adolescents (Table 7).

Regarding the relationship among BMI, SBP and DBP of the study group in different study periods, BMI and SBP show a significant positive correlation at *p* = 0.000 level, proving that BMI was strongly correlated with blood pressure and BMI B.P. also increases (Figure 4). Hence, there is the greatest need for early diagnosis and treatment of overweight to significantly reduce hypertension status and prevent CVD risk factors at the earliest. Related to academic performance, in the post test-2, there was a significant association of academic performance with BMI at *p* = 0.003 among the study group adolescents (Table 8). 

## 4. Discussion

Childhood obesity and hypertension are epidemics that pose a major global challenge to chronic disease prevention and long-term health and preventing them may have the most long-term benefits. This project is a school-based intervention (SEAL-MI) that uses an innovative approach to prevent overweight and hypertension among adolescents through promoting healthy lifestyles. The plan targets families and is implemented via schools. The study’s objectives were to find the correlation between BMI and BP and BMI with academic performance. 

One of the project’s major strengths was that we were able to conduct a 6 months intervention involving the school teachers and family. Another point of strength is the multi-faceted behavioral approach. A systematic review found that Family Based Intervention studies discovered significant changes in two or more behavioral aspects. This study focused on three behavioral domains (PA, dietary habits, and Sleep) directly and depression indirectly. In addition, as this study was focusing on academic performance of the adolescent, there was much response from the parents and teachers and they acted as ‘change agents’.

In the present study, the intervention group had 520 adolescents and the control group had 485 initially, with the dropout rate of 3.85% in the intervention group at the end of the study. Among the 12,895 adolescents screened, 28.73% were found to be overweight, and 9.26% were prehypertensive. Among the overweight adolescents, the prevalence of prehypertension was 32.25%, which was relatively very high. Many studies emphasized the same trend of overweight and hypertension among adolescents [16,17,18,19,20]. This finding is consistent with the ORANGE project by Kumaravel V et al., 2014 at Chennai, which concluded that the prevalence of overweight/obesity was 26.4% among children aged 6–11 years [21]. Regarding the prevalence of hypertension, many studies conducted by various researchers [17,18,19,20,22], including the present study, found that hypertension was significantly high among overweight and obese adolescents. 

In this study, the reduction in the BMI and B.P. of the intervention group adolescents was statistically significant after 3 and 6 months of intervention. The comparison of BMI and B.P. in the study group, done by independent “*t*” test, showed that though it was not significant, the difference found during the study periods indicated that the SEAL-MI was highly effective in reducing the BMI and B.P. It also emphasizes that long-term intervention with appropriate reinforcement is needed to meet the desired change. It also indicates that it is easy to reduce BMI and B.P. initially; sustaining the reduction was the more significant challenge. According to a systematic review, the level of parental involvement appeared to have a positive impact on the effectiveness of interventions on children’s weight and energy balance-related behaviors [23].Another compilation report suggests that interventions encouraging participant engagement may benefit disadvantaged groups more than higher literacy/SES status families [24]. With respect to family income, 50% of the study and 53% control group had the family income of Rs.10, 001–15,000/which comes in lower income households, which may be a possible reason for the improved outcomes of this study.

A study by Lone DK et al., 2014 suggested that a chronic aerobic exercise intervention lasting 4–8 months effectively reduces blood pressure in overweight adolescents at high to moderate intensities [25]. Ghosh S et al., (2010) also concluded that a significant inverse correlation was observed in mean arterial pressure with the duration of walking, cycling, and gym [26]. These findings were consistent with other studies, suggesting that we need to provide more opportunities for overweight and obese adolescents to be active throughout the week, emphasizing physical activity during school hours [27]. Additionally, Sung K D et al., (2019) reported that rope exercise might be an effective intervention to improve the CVD risk factors in prehypertensive adolescent girls. Jumping rope is an easily accessible exercise modality that may have significant health implications for CVD prevention in younger populations [28]. These results highlight the importance of multidisciplinary programs including physical exercise and diet education to prevent and treat childhood hypertension, overweight/obesity such as SEAL-MI and emphasize their encouraging long-term positive effects on CVD risk factors.

Regarding dietary habits, physical activity and sleep, there was a significant difference within the study group at *p* = 0.000 level after six months of intervention. According to Kim J et al., (2020), there were significant improvements following the 12-week exercise program for body fat%, waist circumference, systolic BP, blood glucose, and insulin levels. This study evidenced that rope exercise intervention can be a beneficial therapeutic intervention to improve CVD risk factors in obese adolescent girls with prehypertension [29].

An interesting fact emerged from the study that there was a negative association between the BMI and academic performance of the adolescents during the study periods. It shows that when BMI increases, the academic performance of the adolescents’ declines. A similar finding was reported in a few other studies as well [30,31]. Obesity increases the risk of developing early puberty in adolescents [32], menstrual irregularities in adolescent girls [33], and sleep disorders such as Obstructive Sleep Apnea (OSA) [34,35]. The Excess adiposity may also persuade various aspects of pubertal development, such as pubertal initiation and hormonal parameters during puberty. These alterations may not be harmless. For example, earlier puberty in girls appears to be associated with a higher risk of psychological problems, risk-taking behavior, and even future breast cancer. Obesity during childhood may lead to early signs of puberty (thelarche) in girls and pubertal delay in boys. Girls with obesity are at risk for hyperandrogenemia due to increased total testosterone production and reduced sex hormone-binding globulin (SHBG). Hyperandrogenemia in adolescence may portend adult PCOS and its potential metabolic and cardiovascular complications [36]. Though this aspect is not the scope of this study, obesity in the pubertal stage hampers adolescents’ sexuality and reproductive health. Hence, screening adolescents for obesity and early intervention to prevent these physical and psychosocial problems in pubertal adolescents is mandatory.

In this study, there was a statistically significant association between the demographic variables such as gender, level of education, number of family members, type of family, family history of obesity and chronic illness with BMI and systolic B.P. of the adolescents. The present study results were consistent with other studies [37,38]. The literature survey on the demographic variables associated with BMI has found moderate to high significance [39].

The linear regression analyses revealed that the gender (ß: 0.47, 95% CI: 0.39, 0.81), age (ß: 0.39, 95% CI: 0.17, 0.46), family income (ß: 0.2, 95% CI: 0.41, 0.5), residence (ß: 0.19, 95% CI: 0.01, 0.27), and type of family (ß: 0.25, 95% CI: 0.39, 0.02) had the strongest correlate with the BMI of the adolescents. This suggests that these variables strongly influence adolescents’ BMI status and need to give importance to the adolescents from rural habitats, from poor, nuclear families and the most crucial point of catching them young will bring significant change in the disease burden. Additionally, the mother’s education (ß: 0.35, 95% CI: 0.18, 0.59) had the strongest correlate with the SBP of the adolescents. In contrast, the DBP was negatively persuaded by age (ß: −0.36, 95% CI: 1.54, 0.29) and gender (ß: −0.26, 95% CI: 1.34, 0.12) of the adolescents. Therefore, it is imperative to take these risk factors into consideration in order to enhance the efficacy of preventive actions.

BMI and SBP showed a significant positive correlation in all the three study periods, proving that BMI is strongly correlated with blood pressure and the greatest need for early diagnosis. The treatment of overweight will have a significant reduction in hypertension status. Numerous studies have been performed across the globe on whether or not physical exercise has a significant positive correlation with weight and cardiovascular risk factors [40,41,42,43,44]. These studies have reasonably concluded that there is indeed a positive correlation, although the correlation coefficient varies from study to study between BMI and B.P. The current evidence suggests that exercise intervention with consistent motivation among adolescents improves body composition, mainly by lowering body fat. The limited accessible support further indicates that exercise intervention may improve cardiometabolic risk factors and prevent CVD risk during adulthood. The FIVALIN project concluded that by employing a multi-pronged behavioral approach that targets healthy eating, physical activity, screen time, sleep quality and duration, and psychological well-being, childhood obesity can be prevented in economically deprived families [45]. According to the data, lowering one’s BMI is most closely related to lowering one’s total cholesterol, triglycerides, low density lipoprotein cholesterol, and diastolic and systolic blood pressure [46,47,48].

One of the limitations was, although the current study allowed adolescents’ parents to attend initial education session, it may be beneficial for future similar interventions to involve parents intensively. Evidence suggests that involving families as well as adolescents in a childhood obesity intervention increases the likelihood of intervention success. Another limitation is that no covariates/confounding factors were examined, despite the fact that additional covariates could be useful in investigating individual-level changes.

## 5. Conclusions

The earlier is the best for any preventive strategies. According to the findings of this study, interventions involving lifestyle and behavioral modifications such as physical activity, adaptation of a balanced diet, and improving sleep quality diet, and a longer duration of intervention are effective in the prevention and reduction of childhood obesity and hypertension. Additionally, a decrease in BMI is most closely associated with a decrease in blood pressure. As a result, school-based interventions, such as the weekly after-school clubs with parental supervision at home, may be an effective way to reduce CVD risk factors in adolescents in addition, with a positive bond with their academic performance.

## 6. Implications

Medical experts can feel confident in advising parents to address their at-risk children’s food choices and exercise patterns instantly at the earliest rather than waiting for overweight and the patterns that promote it to resolve on their own. Identifying adolescent obesity allows healthcare professionals to intervene sooner to slow the trajectory of aberrant weight gain that leads to obesity-related illness and CVD risk factors. Preventing these will provide a more significant opportunity for the adolescents to grow to their fullest potential physically, psychologically, socially, and emotionally.

## Figures and Tables

**Figure 1 healthcare-09-01549-f001:**
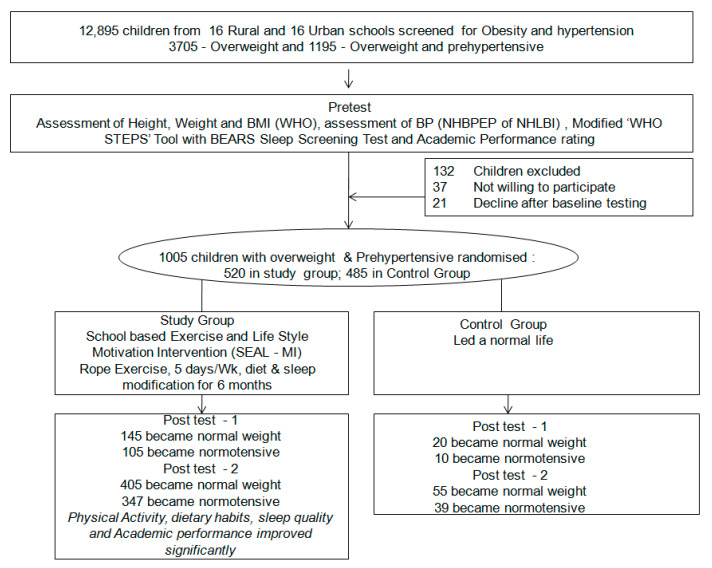
CONSORT flow diagram of the study.

**Figure 2 healthcare-09-01549-f002:**
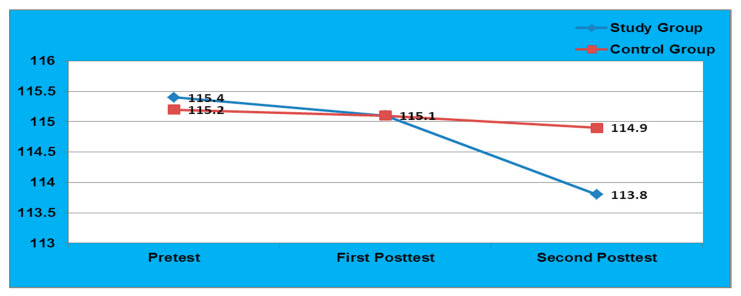
Comparison of SBP among Study and Control Group.

**Figure 3 healthcare-09-01549-f003:**
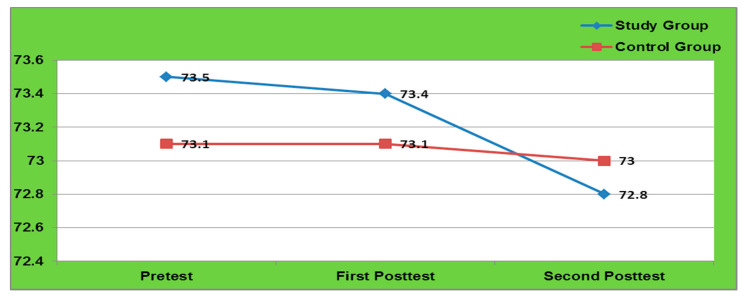
Comparison of DBP among Study and Control Group.

**Figure 4 healthcare-09-01549-f004:**
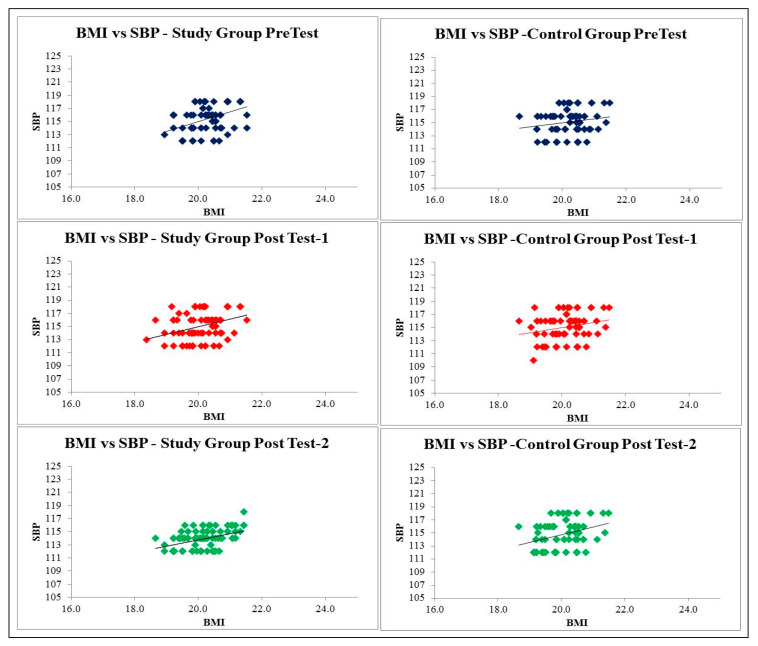
Relationship between BMI and SBP of Study and Control group.

**Table 1 healthcare-09-01549-t001:** Comparison of BMI Status among Study and Control Group.

Study Time	Study Group		Control Group		*t*-Value *p*-Value
	Mean	S.D.	Mean	S.D.	
Pretest	20.3	0.58	20.2	0.51	*t* = 0.416; *p* = 0.675 (NS)
Post test-1	20.1	0.63	20.1	0.55	*t* = 0.474, *p* = 0.635 (NS)
Post test-2	20.1	0.64	20.1	0.57	*t* = 0.257, *p* = 0.797 (NS)
Repeated ANOVA	*F* = 3.619, *p* = 0.036 (S)		*F* = 1.679, *p* = 0.188 (NS)		

S—Significant; NS—Non-significant.

**Table 2 healthcare-09-01549-t002:** Comparison of Effective score for B.P.

Effect Period	Systolic Blood Pressure					Diastolic Blood Pressure				
	Study Group		Control Group		*p*-Value	Study Group		Control Group		*p*-Value
	Mean (S.D.)	*p*-Value	Mean (S.D.)	*p*-Value		Mean (S.D.)	*p*-Value	Mean (S.D.)	*p*-Value	
Pre to Post test-1	0.27 (0.96)	0.006	0.08 (0.57)	0.16	0.10	0.09 (0.34)	0.03	0.03 (0.2)	0.32	0.10
Pre to Post test-2	1.57 (1.53)	0.000	0.31 (1.01)	0.03	0.000	0.69 (1.26)	0.000	0.12 (0.75)	0.11	0.000

**Table 3 healthcare-09-01549-t003:** Comparison of Effective score for Dietary Habits, Physical Activity, and Sleep.

Effective Period	Study Group			Control Group			*p*-Value
	*n*	Mean (S.D.)	*p*-Value	*n*	Mean (S.D.)	*p*-Value	
Dietary Habits							
Pre to Post test-1	500	0.46 (2.7)	0.91	485	2.55 (5.5)	0.91	0.000
Pre to Post test-2	485	6.48 (6.3)	0.000	484	1.95 (5.3)	0.11	0.000
Physical Activity							
Pre to Post test-1	500	18 (9.3)	0.000	485	1.89 (5.3)	0.16	0.000
Pre to Post test-2	485	49.8 (19.5)	0.000	484	2.23 (5.7)	0.21	0.000
BEARS Sleep Score							
Pre to Post test-1	500	3 (7.6)	0.000	485	0.26 (1.8)	0.158	0.000
Pre to Post test-2	485	10.31–14.5	0.000	484	0 (2.6)	1.0	0.000

**Table 4 healthcare-09-01549-t004:** Association between BMI and Systolic B.P. with Demographic Variables of study group adolescents.

S. No.	Demographic Variables	BMI			ANOVA *F* and *p*-Value	Systolic BP			ANOVA *F* and *p*-Value
		No.	Mean	S.D.		No.	Mean	S.D.	
1	Gender								
	Male	226	20.1	0.57	*t* = 2.918 *p* = 0.004	168	118.4	2.08	*F* = 10.91 *p* = 0.000
	Female	274	20.4	0.55		332	117.8	2.01	
2	Education								
	7th Std.	274	20.4	0.6	*F* = 42.293 *p* = 0.000	244	115	2.1	*F* = 11.94 *p* = 0.000
	8th Std.	226	20.6	0.58		256	116	1.4	
3	Number of Family Members								
	≤3	53	20.3	0.42	*F* = 4.430 *p* = 0.006	54	116	1.2	*F* = 4.354 *p* = 0.003
	4	221	20.4	0.57		233	116	2.2	
	5	125	19.9	0.62		66	115	2.4	
	≥6	101	20.3	0.49		147	115	0.9	
5	Type of family								
	Nuclear	178	20.4	0.56	*F* = 3.481 *p* = 0.035	186	116	2.1	*F* = 6.72 *p* = 0.002
	Joint	103	20.1	0.64		113	114	1.8	
	Extended	219	20.1	0.55		201	115	1.6	
6	Family History of Obesity and Chronic Illness								
	Nil	268	20.4	0.47	*F* = 2.952 *p* = 0.036	327	115	2	*F* = 3.083 *p* = 0.031
	DM	127	20	0.75		83	117	2	
	H.T.	72	20.3	0.65		51	116	0	
	Cancer	33	20	0.28		39	118	0	

Nil—No Chronic illness, DM—Diabetes Mellitus, H.T.—Hypertension.

**Table 5 healthcare-09-01549-t005:** Regression Model describing the Effect of BMI on Demographic Variables of Study group.

Variables	PRETEST					POST-TEST-1					POST-TEST-2				
	Beta Coefficient	*t*	*p*	95% CI		Beta Coefficient	*t*	*p*	95% CI		Beta Coefficient	*t*	*p*	95% CI	
				Lower	Upper				Lower	Upper				Lower	Upper
Gender	0.29	3.98	0.000	0.17	0.51	0.34	4.5	0.000	0.24	0.61	0.47	5.71	0.000	0.39	0.81
Religion	0.28	4.21	0.000	0.12	0.32	0.26	3.6	0.001	0.09	0.32	0.05	0.67	0.505	−0.08	0.18
Age	0.67	8.42	0.000	0.37	0.6	0.49	5.88	0.000	0.26	0.52	0.39	4.27	0.000	0.17	0.46
Birth order	−0.12	1.43	0.157	−0.29	0.05	−0.18	2.06	0.042	−0.39	−0.01	−0.17	1.79	0.078	−0.41	0.02
Number of Siblings	0.19	1.88	0.063	−0.01	0.41	0.26	2.06	0.024	0.04	0.53	0.23	1.95	0.054	−0.01	0.54
Family Income (Rs.)	−0.03	0.35	0.728	−0.18	0.12	−0.12	1.46	0.147	−0.34	−0.06	−0.2	2.69	0.009	−0.41	0.05
Residence	0.14	1.83	0.071	−0.01	0.19	0.1	1.27	0.207	−0.04	0.19	0.19	2.18	0.032	0.01	0.27
Type of Family	−0.1	0.98	0.328	−0.22	0.08	−0.16	1.47	0.145	−0.29	0.04	−0.25	2.16	0.03	−0.39	−0.02
	R squared value = 60.6%					R squared value = 53.5%									

**Table 6 healthcare-09-01549-t006:** Regression Analysis of SBP with Demographic Variables among Study Group.

Demographic Variables	PRETEST					POST-TEST-1					POST-TEST-2				
	Beta Coefficient	*t*	*p*	95% CI		Beta Coefficient	*t*	*p*	95% CI		Beta Coefficient	*t*	*p*	95% CI	
				Lower	Upper				Lower	Upper				Lower	Upper
Age	0.52	5.624	0.000	0.88	1.84	0.47	4.77	0.000	0.71	1.73	0.59	5.82	0.000	0.7	1.42
Mother’s Education	0.26	3.05	0.003	0.14	0.68	0.23	2.57	0.012	0.08	0.65	0.35	3.77	0.000	0.18	0.59
Residence	0.32	3.546	0.001	0.33	1.17	0.25	2.58	0.012	0.13	1.01	0.26	2.71	0.008	0.11	0.73
R squared value = 48.4%						R squared value = 44.4%					R squared value = 44.0%				

**Table 7 healthcare-09-01549-t007:** Regression Analysis of DBP with Demographic Variables among Study Group.

Demographic Variables	PRETEST					POST-TEST-1					POST-TEST-2				
	Beta Coefficient	*t*	*p*	95% CI		Beta Coefficient	*t*	*p*	95% CI		Beta Coefficient	*t*	*p*	95% CI	
				Lower	Upper				Lower	Upper				Lower	Upper
Gender	−0.23	2.13	0.036	−1.86	−0.06	−0.24	2.21	0.029	−1.91	−0.1	−0.26	2.38	0.020	−1.34	−0.12
Age	−0.36	2.87	0.005	−2.19	−0.4	−0.31	2.46	0.016	−2.03	−0.21	−0.36	2.91	0.005	−1.54	−0.29
R squared value = 15.1%						R squared value = 20.2%									

**Table 8 healthcare-09-01549-t008:** Relationship among BMI, SBP, and DBP of study and control group.

Study Periods	Correlation Variables	Study Group		Control Group	
		*r*-Value	*p*-Value	*r*-Value	*p*-Value
Pretest	BMI and SBP	0.412	0.000	0.169	0.099
	BMI and DBP	0.01	0.918	0.004	0.973
	BMI and Academic Performance	−0.006	0.953	−0.102	0.318
Post test-1	BMI and SBP	0.355	0.000	0.0222	0.29
	BMI and DBP	0.052	0.607	0.005	0.959
	BMI and Academic Performance	−0.087	0.391	−0.17	0.095
Post test-2	BMI and SBP	0.421	0.000	0.0339	0.30
	BMI and DBP	0.134	0.191	0.002	0.982
	BMI and Academic Performance	−0.301	0.003	−0.193	0.058

## Data Availability

The data presented in this study areavailable on request from the corresponding author.

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
