# Peer review of "School-Based Exercise and Life Style Motivation Intervention (SEAL.MI) on Adolescent’s Cardiovascular Risk Factors and Academic Performance: Catch Them Young"

_healthcare, 2021, doi:10.3390/healthcare9111549_

Round 1

Reviewer 1 Report

Introduction

There is no better explanation for why life style motivation is applied. The contents of lines 93-98 are not closely related to this study

Research methods

Is the teacher who guides the student movement from the student's school or assigned by the research group?

Has the instructor received intensive training?

There is no specific solution for life style motivation.

Is the manual mercury sphygmomanometer or electronic sphygmomanometer used for blood pressure test?

What questionnaire is used for sleep evaluation?

Statistical methods suggest the use of covariance analysis with previous measured data as covariates.

result

There was no demographic data of the subjects.

What is the gender composition of each group? Initial weight, height, BMI, etc.

Research methods

Is the teacher who guides the student movement from the student's school or assigned by the research group?

Has the instructor received intensive training?

There is no specific solution for life style motivation.

Is the manual mercury sphygmomanometer or electronic sphygmomanometer used for blood pressure test?

What questionnaire is used for sleep evaluation?

Statistical methods suggest the use of covariance analysis with previous measured data as covariates.

result

There was no demographic data of the subjects.

What is the gender composition of each group? Initial weight, height, BMI, etc.

Author Response

Introduction

  • There is no better explanation for why life style motivation is applied. The contents of lines 93-98 are not closely related to this study

Included in Line:72 -77

 Research methods

  • Is the teacher who guides the student movement from the student's school or assigned by the research group?

      Changes made and highlighted

      Line: 186 -189

  • Has the instructor received intensive training?

      Changes made and highlighted

      Line: 186 -189

  • There is no specific solution for life style motivation.

      Changes made and highlighted

Line : 179 -182

The school-based Exercise and Life Style Motivation Intervention (SEAL - MI) incorporated planned teaching on knowledge regarding prevention and control of blood pressure, overweight, dietary modifications, the rope exercise program and activities to enhance sleep quality. Cumulatively, these interventions brought out the desired changes in the study variables.

  • Is the manual mercury sphygmomanometer or electronic sphygmomanometer used for blood pressure test?

      Changes made and highlighted

Line: 210- 214

  • What questionnaire is used for sleep evaluation?

      Changes made and highlighted

Line: 133 – 134

  • Statistical methods suggest the use of covariance analysis with previous measured data as covariates

 In the present study, ANCOVA was not done.

 Result

There was no demographic data of the subjects.

  • What is the gender composition of each group?

Considering the gender, approximately equal number of the sample, 231(44.4%) boys and 285 (55.6%) girls in study as well as 281(57.9%) girls and 204(42.06%) boys in the control group participated in the study

Line: 247 – 249

  • Initial weight, height, BMI, etc.

It is given in table as pretest data

Line: 260

Reviewer 2 Report

ACCEPT WITH SIGNIFICANT MODIFICATIONS

 Interesting research is presented. However, for publication, you should: Improve the wording of the entire article. Especially, in: INTRODUCTION, DISCUSSION and CONCLUSIONS.

INTRODUCTION: Sections not related to a coherent exhibition are made. It does not provide an adequate view of the status of the problem to be investigated. IT NEEDS TO BE RECONSIDERED.

METHOD: DESIGN:The type of design used must be specified more precisely. POPULATION AND ENVIRONMENT: More must be specified. It is very incomplete. INSTRUMENTS: It must be specified with greater academic rigor.

DISCUSSION: It must be totally reconsidered.

CONCLUSIONS: It must complete and relate it to the results obtained.

Author Response

Interesting research is presented. However, for publication, you should: Improve the wording of the entire article. Especially, in: INTRODUCTION, DISCUSSION and CONCLUSIONS.

INTRODUCTION: Sections not related to a coherent exhibition are made. It does not provide an adequate view of the status of the problem to be investigated. IT NEEDS TO BE RECONSIDERED.

  • Changes made and highlighted

METHOD:  DESIGN:

The type of design used must be specified more precisely.

  • Changes made and highlighted

Line 83

POPULATION AND ENVIRONMENT:

More must be specified. It is very incomplete.

  • Changes made and highlighted

Line: 88 -90

INSTRUMENTS: It must be specified with greater academic rigor.

  • Changes made and highlighted

Line: 122 – 177

DISCUSSION: It must be totally reconsidered.

  • Changes made and highlighted

 CONCLUSIONS: It must complete and relate it to the results obtained.

  • Changes made and highlighted

Reviewer 3 Report

First of all, I would like to congratulate the authors for the work presented.
The subject matter investigated is interesting and somewhat original, and certainly necessary, since cardiovascular risk is not exactly an unimportant variable.

The table of contents provides the data necessary to understand the work as a whole.

The introduction provides the bibliographic references that support the research, so they can be considered correct; it would be interesting to include some more references, corresponding to the year in force, 2021 (there is only one reference).

Regarding the methodology used, the calculation of the power of the sample, which is very good, is to be appreciated. The work program is explained correctly and in detail, very well.

The research design, is it an experimental or quasi-experimental...all extraneous variables have been controlled...it would be good to clarify this.

The flow chart makes the study quite clear.

Regarding the statistical analysis, it would be good for the authors to clarify and justify why each one. With this, for example, a bivariate analysis is done (Pearson's or Spearman's correlation?), how is the data distribution, is it normal or not. Chi-square why, to which variables?..., if ANOVA is used a normal data distribution is assumed, with which test has been performed. In addition a regression analysis is done, why?...with this, the idea is not to criticize the authors, on the contrary, it seems to me a good work, and it would be good to clarify all this.
One last doubt, how is the significance of pretest and posttest justified, is student's t-test applied because there is normality of data?....

In conclusion, I congratulate the authors for their article, I hope it comes to fruition. Best regards.

Author Response

First of all, I would like to congratulate the authors for the work presented.
The subject matter investigated is interesting and somewhat original, and certainly necessary, since cardiovascular risk is not exactly an unimportant variable.

The table of contents provides the data necessary to understand the work as a whole.

The introduction provides the bibliographic references that support the research, so they can be considered correct; it would be interesting to include some more references, corresponding to the year in force, 2021 (there is only one reference).

  • Reference 43-46 included

Line: 583- 597

Regarding the methodology used, the calculation of the power of the sample, which is very good, is to be appreciated. The work program is explained correctly and in detail, very well.

The research design is it an experimental or quasi-experimental...all extraneous variables have been controlled...it would be good to clarify this.

  • The research design is it an experimental or quasi-experimental...

Line: 85 -86

Experimental study with before and after control group design

  • All extraneous variables have been controlled.

Line: 250 – 251

There was homogeneity among selected samples in both the groups tested through “Test of Goodness of Fit”.

The flow chart makes the study quite clear.

Regarding the statistical analysis, it would be good for the authors to clarify and justify why each one. With this, for example, a bivariate analysis is done (Pearson's or Spearman's correlation?), how is the data distribution, is it normal or not. Chi-square why, to which variables?..., if ANOVA is used a normal data distribution is assumed, with which test has been performed. In addition a regression analysis is done, why?...with this, the idea is not to criticize the authors, on the contrary, it seems to me a good work, and it would be good to clarify all this.

  • Changes made

One last doubt, how is the significance of pretest and posttest justified, is student's t-test applied because there is normality of data?....

  • Yes

In conclusion, I congratulate the authors for their article, I hope it comes to fruition. Best regards.

Round 2

Reviewer 1 Report

Accept in present form

Reviewer 2 Report

It is estimated that the above suggestions have been considered. This latest version can be published.